# Systematic review and meta-analysis of the association between peripheral inflammatory cytokines and generalised anxiety disorder

Harry Costello, Rebecca L Gould, Esha Abrol, Robert Howard

Division of Psychiatry, University College London, London, UK

**Correspondence to**
Dr Harry Costello;
harry.costello@ucl.ac.uk

## ABSTRACT

**Objective** Inflammation has been implicated in the aetiology of mental illness. We conducted the first systematic review and meta-analysis of the association between peripheral markers of inflammation and generalised anxiety disorder (GAD).

**Design** Systematic review and meta-analysis of studies measuring peripheral cytokine levels in people with GAD compared with controls.

**Data sources** MEDLINE (1950–), EMBASE (1947–), PsycINFO (1872–) and Web of Science (1945–) databases up until January 2018.

**Eligibility criteria** Primary, quantitative research studies of people with a diagnosis of GAD assessed using a standardised clinical interview that measured peripheral inflammatory markers.

**Data extraction and synthesis** Two independent reviewers extracted data and assessed study quality. Meta-analysis using a random-effects model was conducted for individual cytokines where data from three or more studies were available.

**Results** 14 of 1718 identified studies met the inclusion criteria, comprising 1188 patients with GAD and 10 623 controls. In total 16 cytokines were evaluated. Significantly raised levels of C reactive protein (CRP), interferon-γ and tumour necrosis factor-α were reported in patients with GAD compared with controls in two or more studies. Ten further proinflammatory cytokines were reported to be significantly raised in GAD in at least one study. However, 5 of 14 studies found no difference in the levels of at least one cytokine. Only CRP studies reported sufficient data for meta-analysis. CRP was significantly higher in people with GAD compared with controls, with a small effect size (Cohen's d=0.38, 0.06–0.69), comparable with that reported in schizophrenia. However, heterogeneity was high (I²=75%), in keeping with meta-analyses of inflammation in other psychiatric conditions and reflecting differences in participant medication use, comorbid depression and cytokine sampling methodology.

**Conclusion** There is preliminary evidence to suggest an inflammatory response in GAD, but it remains unclear whether inflammatory cytokines play a role in the aetiology. GAD remains a poorly studied area of neuroinflammation compared with other mental disorders, and further longitudinal studies are required.

### Strengths and limitations of this study

► This is the first study to conduct a comprehensive systematic review and meta-analysis of peripheral inflammatory markers in generalised anxiety disorder.

► A wide range of databases were searched and a large number of papers screened for inclusion in the study, 14 of which were subjected to quality assessment and detailed critical appraisal.

► It was only possible to conduct a meta-analysis of C reactive protein, and it was not possible to examine publication bias due to the limited number of studies identified for inclusion in the meta-analysis.

► The high levels of heterogeneity across studies mean that findings should be interpreted with caution.

## INTRODUCTION

There is growing evidence for immune-mediated pathogenic mechanisms in several psychiatric disorders with discrete profiles of inflammatory mechanisms.[1] Epidemiological evidence has shown an increased risk of mood disorders and psychosis in people with a history of severe infection or autoimmune conditions.[2 3] This has been supported by genome-wide association studies implicating multiple immune signalling pathways,[4] and altered profiles of proinflammatory cytokines and acute phase reactants in schizophrenia,[5] depression,[6] obsessive compulsive disorder (OCD)[7] and bipolar disorder.[8] However, the relationship between inflammation and mental illness remains poorly understood and controversial, with a number of proposed potential neuropathological mechanisms,[9 10] including changes in microglial function,[1] glutamatergic excitotoxicity,[11] synaptic plasticity[12] and reduced hippocampal neurogenesis.[13]

Despite increasing interest in the role of inflammation in mental illness, relatively little research has focused on potential associations

with anxiety disorders.[14] These are common, with an estimated lifetime prevalence of 7.3%–28.8%, are associated with substantial functional impairment and are estimated to cost between $42 and $47 billion to the US economy each year.[15 16] However, only 60% of patients are thought to respond to pharmacological and psychological treatments, and understanding of the underlying pathophysiological mechanisms of anxiety disorders remains poor.[17]

Generalised anxiety disorder (GAD) is the most common anxiety disorder, with a degree of associated disability equivalent to that of major depressive disorder (MDD).[18] Despite psychopharmacological[19] and psychological[20] treatments showing effectiveness in GAD, 42% of people living with GAD experience ongoing symptoms after 12 years, and half of remitted patients experience recurrence.[18] GAD is more prevalent in those with inflammatory conditions such as rheumatoid arthritis (RA),[21 22] with case series studies suggesting symptoms are less common with immune-modulating treatment targeting specific inflammatory cytokines.[23] The chronic clinical course and relatively high probability of recurrence in GAD, in addition to preliminary evidence of an inflammatory component in other anxiety disorders,[7 24] suggest that inflammation could be an important neurobiological mechanism in the aetiology of this disorder.

To date, two previous reviews of inflammatory biomarkers in GAD have been conducted. Of these, however, one was a narrative review[25] and the other was restricted to literature published within the last decade,[14] and with a focus on all anxiety disorders. Both reviews reported that there was preliminary evidence for inflammatory changes in GAD. However, only three studies were identified by the systematic review reporting cytokine changes in GAD and no meta-analysis was performed. No study to date has conducted a comprehensive systematic review and meta-analysis of all current literature focusing on GAD or commented on the longitudinal association between inflammation and GAD.

We aimed to systematically review the cross-sectional and longitudinal associations between inflammatory biomarkers and GAD, and perform the first meta-analysis of inflammatory biomarkers in GAD.

## METHODS

We conducted a systematic review of studies that had included people with GAD who had undergone peripheral cytokines measurement and a between-group meta-analysis of cytokine levels in people with GAD compared with controls. We conducted the study according to the Preferred Reporting Items for Systematic Reviews and Meta-Analyses guidelines.[26]

We searched MEDLINE (1950–), EMBASE (1947–), PsycINFO (1872–) and Web of Science (1945–) databases up until January 2018. Reference lists of eligible studies were then searched for further ones that met the eligibility criteria.

Our search terms (see online supplementary appendix 1 for further details) were the following: (inflammat* or cytokine or interferon or IFN or interleukin or 'translocator protein' or TSPO or 'tumour necrosis factor' or 'tumor necrosis factor' or TNF or IL-1 or IL-2 or IL-4 or IL-7 or IL-6 or IL-8 or IL-10 or migroglia or t-cell or lymphocyte or 'C-reactive protein' or 'C reactive protein' or CRP or 'acute phase protein' or 'fibrinogen') and ('generalised anxiety disorder' or 'generalized anxiety disorder' or GAD or worry).

We included primary, quantitative research studies (including unpublished theses and dissertations), written in any language, that included people with a diagnosis of GAD assessed using standardised clinical interview (eg, Structured Clinical Interview for DSM[27]) or standardised psychometric instruments. Studies reported cross-sectional or longitudinal data in clinical or community populations. Cross-sectional studies measured inflammatory biomarker concentrations in anxious people versus non-anxious healthy controls, while longitudinal studies measured inflammatory biomarker concentrations at baseline and anxiety scores at follow-up. Inflammatory markers were measured in the unstimulated state (no antigen-induced stimulation of cytokine production) and sampled from peripheral blood, cerebrospinal fluid (CSF) or saliva at any time of day. Exclusion criteria included studies with less than five participants, studies on animals and studies where subjects were participants in the treatment arms of clinical trials.

### Patient and public involvement

There was no patient or public involvement in the study.

### Data extraction and quality assessment

Data were extracted and quality assessed for all studies that met the eligibility criteria by two independent raters (HC, EA), with disagreements settled by consensus and discussion. For each cytokine, we extracted the means, variance estimates or 95% CIs and sample size for GAD and control groups. We also extracted demographic data (eg, age, sex) and clinical data (eg, medication use, comorbid depression, severity) where available. Authors were contacted for further information, where necessary.

Risk of bias and study quality were evaluated using the Newcastle-Ottawa Quality Assessment Scale.[28] Other potential confounding factors (including assay type and sensitivity, inflammatory marker analysis and recruitment methods) were also examined to allow more detailed bias and quality analysis of studies.

### Strategy for data synthesis

Separate meta-analyses were performed for individual biomarkers in GAD versus controls if sufficient data were available from a minimum of three studies. Due to different measurement methods and anticipated high heterogeneity, we estimated a standardised mean difference (SMD) for each inflammatory marker and used a random-effects model for meta-analysis, conducted using

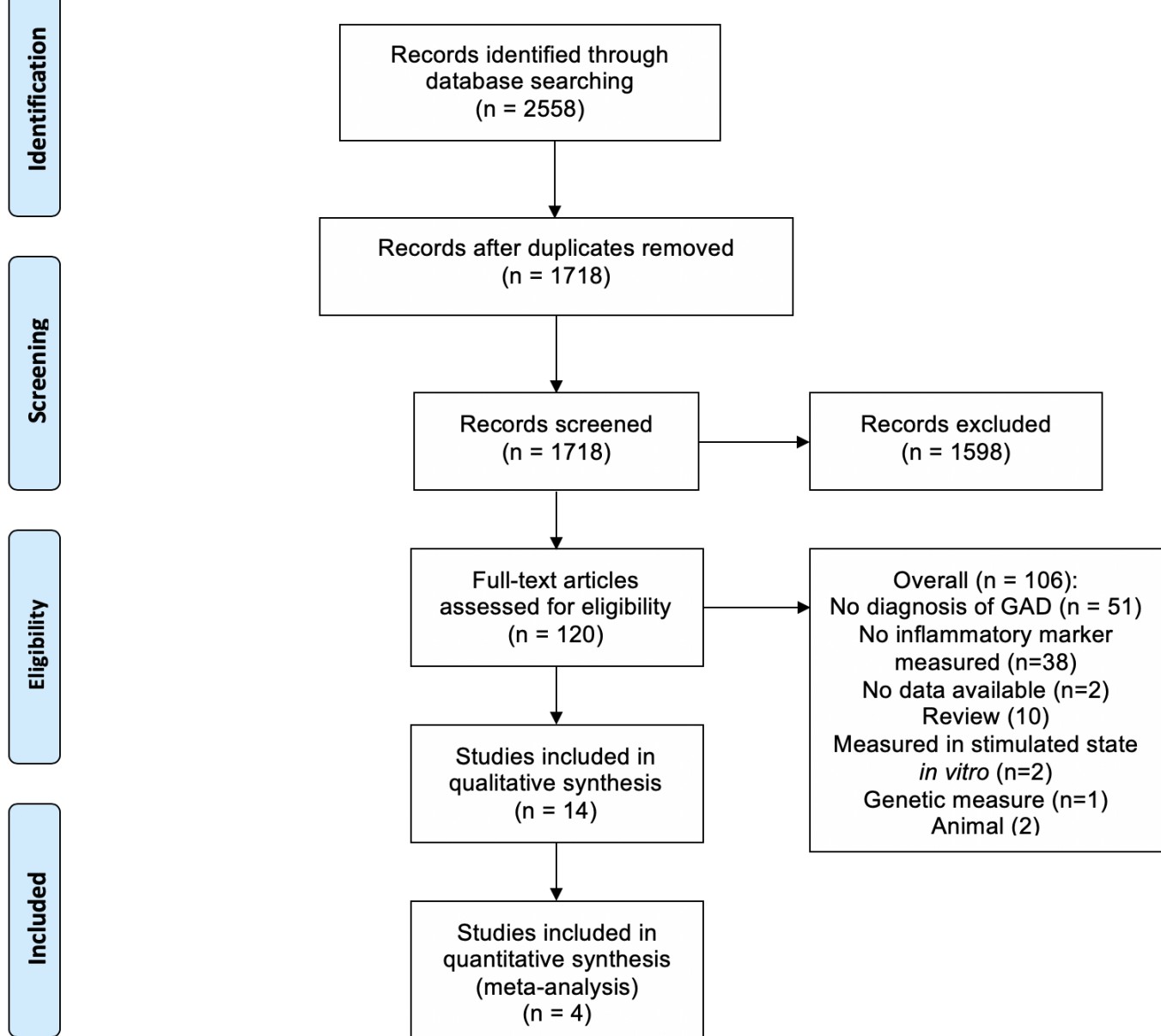

**Figure 1** Flow of studies in the systematic review and meta-analysis. GAD, generalised anxiety disorder.

RevMan V.5. Heterogeneity across studies was quantified with the $I^2$ statistic, with a value of 25% typically regarded as low, 50% as medium and 75% as high.[29] If studies were longitudinal or trials of interventions with multiple data collection points, we examined baseline data only to avoid skewed meta-analysis from inclusion of more than one effect size from the same study.

## RESULTS
### Systematic review
We identified 1718 papers, excluded 1598 of these by titles and abstracts, and retrieved the remaining 120 papers, of which 14 met the eligibility criteria and were included in the final systematic review (see figure 1).

The primary reasons for rejection were that no diagnosis of GAD was recorded or no inflammatory marker was measured.

The characteristics of the 14 included studies are shown in tables 1 and 2. Studies comprised a total of 1188 people with a diagnosis of GAD and 10 623 controls, with a further 116 participants from a study that did not report GAD and control group sizes.

In total, 16 different cytokines were evaluated (see table 2). C reactive protein (CRP) (9/14 studies, 64.2%), tumour necrosis factor-α (TNF-α) (6/14 studies, 42.9%), interleukin-6 (IL-6) (5/14 studies, 35.7%) and interferon-γ (IFN-γ) (3/14 studies, 21.4%) were the most commonly studied. All other cytokines were only analysed in two or less studies.

**Table 1** Study and clinical characteristics

| Study | Country | Study type | Inflammatory markers | N GAD | N Control | Standardised diagnostic assessment/crit era | Anxiety measure | Age (years, SD) GAD | Age (years, SD) Control | % Female GAD | % Female Control | BMI GAD | BMI Control | Current smoking | Medicated | Physical health comorbidities | Mental health comorbidities |
|---|---|---|---|---|---|---|---|---|---|---|---|---|---|---|---|---|---|
| Bankier et al[38] | USA | Case-control | CRP | 15 | 30 | SCID, DSM-IV | NR | NR | 67.6 (12.7) | NR | 33% | NR | NR | GAD: NR Controls: 10% | NR | All participants had CVD. Excluded other conditions. | Excluded. |
| Copeland et al[39] | USA | Cohort, prospective | CRP | 146 | 5664 | Child and Adolescent Psychiatric Assessment <16, Young Adult Psychiatric Assessment >16, DSM-IV criteria | Total number of anxiety symptoms (range: 0–6) | 14.21, all subjects (odds or means ratio 1.12 (1.03–1.22) with GAD dx) | – | 48.7% (odds or means ratio 2.02 (1.04–3.92) with GAD dx) | – | 22.37 (5.62) (odds or means ratio 1.08 (1.05–1.11) with GAD dx) | – | Total sample: 13.5% (odds or means ratio with GAD: 2.86) | 30.2% 'use medication' (odds/means ratio with GAD: 2.00) | 34.7% had 'recent health ailments'. | Total sample: 39.9% comorbid MDD. |
| De Berardis et al[42] | Canada | Cross-sectional | CRP | 70 | No control | SCID, DSM-IV | HAM-A (score >20 for inclusion) | 28.2 (5.3) | – | 51.40% | – | 22.1 (1.67) | – | NR | Excluded. | Excluded. | Total sample: 44.3% of participants had alexithymia. Excluded other comorbid mental illness. |
| Hoge et al[44] | USA | RCT | TNF-α, IL-6 | 70 | No control | SCID, DSM-IV | NR | 39.12 | – | 45.70% | – | NR | – | NR | Excluded. | Excluded. | GAD: 14.3% comorbid MDD. Excluded other comorbid mental illness. |
| Hou et al[47] | UK | Case-control | IL-4, IL-10, TNF-α, IFN-γ | 54 | 64 | MINI, DSM-IV, ICD-10 criteria | HADS, GAD-7 (score >10 for inclusion) | 35.06 (14.45) | 25.75 (8.87) | 34% | 50% | 24.84 (5.70) | 22.45 (3.27) | GAD: 22% Controls: 34% | GAD: 67% use 'anxiolytic' medication. Excluded other medication use. | Excluded. | Excluded. |
| Khandaker et al[33] | UK | Cohort, prospective | CRP | 26 | 3392 | DAWBA, DSM-IV criteria | DAWBA | 15.56 (0.24) | 15.53 (0.31) | 90% | 52.30% | 22.55 (3.52) | 21.40 (3.63) | NR | NR | NR | GAD: 30.77% comorbid MDD. Excluded other comorbid mental illness. |
| Korkeila et al[40] | Finland | Cross-sectional | CRP, TNF-α, IFN-γ | 116 | | MINI | NR | NR | NR | 100% | 100% | All 25.3 (5.0) | | Excluded. | NR | NR | NR |
| Nayek and Ghosh[34] | India | Case-control | CRP | 50 | 50 | ICD-10 | NR | 37.96 (10.7) | 37.00 (12.08) | 54% | 23% | NR | NR | Excluded. | Excluded if using HRT or OCP. Other medications not reported. | Excluded. | NR |
| Ogłodek et al[48] | Poland | Case-control | SDF-1, CCL-5, MCP-1 | 120 | 40 | DSM-V | NR | 41.4 (3.5) | 40.8 (3.1) | 50% | NR | NR | NR | NR | Excluded. | Excluded. | All participants had comorbid personality disorder. Excluded other comorbid mental illness. |

Continued

**Table 1** Continued

| Study | Country | Study type | Inflammatory markers | N GAD | N Control | Standardised diagnostic assessment/crit era | Anxiety measure | Age (years, SD) GAD | Age (years, SD) Control | % Female GAD | % Female Control | BMI GAD | BMI Control | Current smoking | Medicated | Physical health comorbidities | Mental health comorbidities |
|---|---|---|---|---|---|---|---|---|---|---|---|---|---|---|---|---|---|
| Tang et al[35] | China | Case-control | CRP, IL-1α, IL-2, IL-5, IL-6, IL-8, IL-12p70, IFN-γ, GM-CSF | 48 | 48 | MINI, DSM-IV | GAD-7, SAI, TAI | 40.75 (12.21) | 39.56 (10.06) | 58.33% | 64.17% | 22.56 (2.73) | 22.69 (2.63) | GAD: 29% Controls: 23% | Excluded. | Excluded acute illness. Chronic comorbidities NR. | Excluded. |
| Tofani et al[46] | Italy | Case-control | IL-2, IL-10 | 14 | 10 | MINI, DSM-IV | GAD-7 | NR | NR | NR | NR | NR | NR | NR | Excluded. | NR | Excluded. |
| Vogelzangs et al[36] | Holland | Cohort | CRP, IL-6, TNF-α | 454 | 556 | CIDI, DSM criteria | BAI | Total sample: 41.8 (13.1) | | 66.90% | | 25.6 (5.1) | | Total sample: 38.2% | NR | Total sample: 6.2% CVD, 4.9% diabetes, mean of 0.4 other chronic diseases. | Total sample: 58.4% comorbid MDD. Excluded other comorbid mental illness. |
| Yang et al[45] | China | Case-control | IL-1, IL-4, TNF-α | 28 | 41 | MINI, DSM-IV | HAM-A | 55.1 (6.9) | 55.9 (5.6) | 53.60% | 48.80% | 22.0 (4.4) | 22.5 (3.7) | GAD: 35.7% Control: 24.4% | Excluded. | Additional group with comorbid asthma. Excluded other comorbidities. | Excluded. |
| Zahm[41] | USA | Cohort, prospective | CRP, IL-6, TNF-α | 93 | 728 | CDIS, DSM-IV | NR | 68 (9.6) | NR | 17% | NR | NR | NR | NR | NR | All patients had history of CVD. | GAD: 60.0% comorbid MDD, 86.2% had lifetime history MDD. |

–, not applicable; BAI, Beck Anxiety Inventory; CCL-5, chemokine C-C motif ligand 5; CDIS, Computerized Diagnostic Interview Schedule; CIDI, Composite Interview Diagnostic Instrument; CRP, C reactive protein; CVD, cardiovascular disease; DAWBA, Development and Well-Being Assessment; DSM, Diagnostic and Statistical Manual of Mental Disorders; dx, diagnosis; GAD, generalised anxiety disorder; GAD-7, Generalised Anxiety Disorder Assessment; GM-CSF, granulocyte-macrophage colony-stimulating factor; HADS, Hospital Anxiety and Depression Scale; HAM-A, Hamilton Anxiety and Depression Scale; HRT, hormone replacement therapy; ICD, International Classification of Diseases; IFN-γ, interferon-γ; IL, interleukin; MCP-1, monocyte chemoattractant protein-1; MDD, major depressive disorder; MINI, Mini-International Neuropsychiatric Interview; NR, not reported; OCP, oral contraceptive pill; RCT, randomised controlled trial; SAI, State Anxiety Inventory; SCID, Structured Clinical Interview for DSM-IV; SDF-1, stromal derived factor-1; TAI, Test Anxiety Inventory; TNF-α, tumour necrosis factor-α.

**Table 2** Inflammatory marker sampling and analysis

| Study | GAD | Control | Inflammatory markers | Nature of sample | Cross-sectional or longitudinal | Time of day of sample | Fasted period before sample | Assay method | Assay sensitivity reported | Inflammatory marker cut-off used | Confounding factors controlled for |
|---|---|---|---|---|---|---|---|---|---|---|---|
| Bankier et al[38] | 15 | 30 | CRP | Blood | Cross-sectional | NR | NR | High sensitivity turbidimetric immunoassay | Yes | CRP >3mg/L for significance | Age, sex, education, MDD, obesity, smoking history, type II diabetes mellitus, hypertension, hyperlipidaemia, other mental illness. |
| Copeland et al[39] | 146 | 5664 | CRP | Whole blood spots | Longitudinal: sampled aged 9–16, 19 and 21years old | NR | NR | Biotin-streptavidin-based immunofluorometric system | Yes | Excluded if >10mg/L | Age, sex, race, SES, BMI, medication use, substance use, recent physical illness, chronic illness. |
| De Berardis et al[42] | 70 | No control | CRP | Serum | Cross-sectional | 07:00–08:30 | 10-hour fast | Highly sensitive nephelometric assay | Yes | No | Age, sex, BMI, MDD, physical illness, other mental illness, medication use |
| Hoge et al[44] | 70 | No control | TNF-α, IL-6 | Plasma | Longitudinal: sampled prepsychological and postpsychological intervention | 13:00–16:30 | NR | NR | No | No | Age, sex, ethnicity, MDD, medication use, physical illness, other mental illness |
| Hou et al[47] | 54 | 64 | IL-4, IL-10, TNF-α, IFN-γ | Serum | Cross-sectional | 09:00–10:00 | NR | Multiplex ultrasensitive immunoassay | Yes | No | Age, sex, BMI, smoking, alcohol consumption, MDD, physical illness, other mental illness. |
| Khandaker et al[33] | 26 | 3392 | CRP | Serum | Cross-sectional | NR | 'Overnight' | Automated particle-enhanced immunoturbidimetric assay | No | Excluded if >10mg/L | Age, sex, parental SES, ethnicity, maternal age at delivery, concurrent infection, family history of inflammatory disease, MDD. |
| Korkeila et al[40] | 116 | | CRP, TNF-α, IFN-γ | Blood | Cross-sectional | NR | NR | NR | No | No | BMI. |
| Nayek and Ghosh[34] | 50 | 50 | CRP | Serum | Cross-sectional | NR | NR | Particle-enhanced turbidimetric immunoassay technique | No | Excluded if 'raised ESR' | Age, sex, SES, religion, marital status, locality, BMI >30, physical illness. |

**Table 2** Continued

| Study | GAD | Control | Inflammatory markers | Nature of sample | Cross-sectional or longitudinal | Time of day of sample | Fasted period before sample | Assay method | Assay sensitivity reported | Inflammatory marker cut-off used | Confounding factors controlled for |
|---|---|---|---|---|---|---|---|---|---|---|---|
| Ogłodek et al[48] | 120 | 40 | SDF-1, CCL-5, MCP-1 | Plasma | Cross-sectional | 07:00–09:00 | Fasted, duration NR | ELISA | Yes | No | Sex, other mental illness, physical illness, substance misuse, smoking status, medication use. |
| Tang et al[35] | 48 | 48 | CRP, IL-1α, IL-2, IL-5, IL-6, IL-8, IL-12p70, IFN-γ, GM-CSF | Serum | Cross-sectional | 09:00–10:00 | NR | ELISA | No | No | Age, sex, education, BMI, smoking status, alcohol consumption, acute physical illness, other mental illness, medication use. |
| Tofani et al[46] | 14 | 10 | IL-2, IL-10 | Plasma | Cross-sectional | NR | NR | Immunoenzymatic assay | No | No | Medication use. |
| Vogelzangs et al[36] | 454 | 556 | CRP, IL-6, TNF-α | Plasma | Cross-sectional | 08:00–09:00 | 'Overnight' | ELISA | Yes | No | Age, sex, education, smoking status, alcohol intake, physical activity, BMI, physical illness, medication use, MDD, other mental illness. |
| Yang et al[45] | 28 | 41 | IL-1, IL-4, IL-6, TNF-α | Saliva | Cross-sectional | NR | 'Overnight' | ELISA | Yes | No | Age, sex, smoking status, BMI, medication use, physical illness, other mental illness. |
| Zahm[41] | 93 | 728 | CRP, IL-6, TNF-α | Serum | Cross-sectional | No (fasting, duration NR) | Fasted, duration NR | ELISA | Yes | No | Age, sex, SES, BMI, illicit substance use, alcohol use, smoking status, physical activity, physical illness. |

BMI, body mass index; CCL-5, chemokine C-C motif ligand 5; CRP, C reactive protein; ESR, erythrocyte sedimentation rate; GAD, generalised anxiety disorder; GM-CSF, granulocyte-macrophage colony-stimulating factor; IFN-γ, interferon-γ; IL, interleukin; MCP-1, monocyte chemoattractant protein-1; MDD, major depressive disorder; NR, not reported; SDF-1, stromal derived factor-1; SES, socioeconomic status; TNF-α, tumour necrosis factor-α.

Twelve studies (85.7%) reported the assay method used, all of which were versions of an ELISA or enzyme immunoassay. However, only seven studies (50%) reported assay sensitivity. All but one study used blood component samples to assess inflammatory marker levels, with the most common sample type being serum (n=6, 42.9%) and plasma (n=4, 28.6%).

### Risk of bias and quality in individual studies

All included studies had adequate case definition, with participants meeting the diagnostic criteria for GAD according to the Diagnostic and Statistical Manual of Mental Disorders (DSM) or the International Classification of Diseases, with 12 studies (85.7%) using a structured clinical interview for assessment (see table 1).

Most (71.4%) studies included people aged 18–65, but two studies (14.3%) only included participants over the age of 50 and a further two studies (14.3%) used adolescent participant cohorts. The majority (78.6%) of studies accounted for age and sex differences in their analyses. Only 8 of 14 (57.1%) studies recorded participants' body mass index (BMI), which is known to correlate with inflammation, and only half of these accounted for BMI differences in analysis of group differences[30] (see table 2).

Use of psychotropic medication and the presence of comorbid MDD are important moderators of inflammation in other psychiatric disorders.[24] Six studies (42.8%) excluded patients who used psychiatric or other immune-modulating medication, although only two studies (4.3%) reported medication use. The majority of studies (64.2%) either excluded patients with comorbid MDD or adjusted for this in the analyses.

Concurrent physical illness is clearly an important determinant of inflammatory cytokine levels, and this was accounted for by the majority of included studies by either excluding participants with comorbidities (5 studies, 35.7%) or adjusting for chronic physical illness in group comparisons (6 studies, 42.8%), although two studies specifically only included participants with comorbid cardiovascular disease (CVD). Use of a predetermined cut-off value for cytokine levels was employed by three studies (21.4%) to ensure that cases with acute infection were excluded from the sample.

Many inflammatory markers exhibit a diurnal pattern of expression and are affected by consumption of food; thus, time of day of sampling and whether the sample was taken in a fasted state are important factors to consider in analysing relative levels of cytokines.[31] However, time of day of sampling was only recorded in a minority of studies (6 studies, 42.8%), and the same number of studies recorded whether fasted samples were taken.

The overall quality of studies included in the review varied significantly, with Newcastle-Ottawa Scale scores ranging from 2 to 9 (see table 3). The area in which most studies were inadequate was in reporting non-response rate and detailing recruitment methods (see table 3). Lowest quality studies were abstracts or dissertations, and

two studies lacked control groups as only patients with GAD were sampled (see table 4).

### C reactive protein

CRP is a critical early proinflammatory surveillance molecule involved in the activation of the complement system and both innate and adaptive immune systems.[32] We identified nine studies that investigated the association between GAD and CRP, comprising a total of 11 486 participants (see table 5).

Four studies, involving 578 patients with GAD and 4046 controls, provided sufficient information to conduct a meta-analysis of CRP levels in GAD[33–36] (see figure 2). This was the only inflammatory marker for which meta-analysis was possible. Meta-analysis showed significantly raised CRP in GAD compared with controls (SMD 0.38, 95% CI 0.06 to 0.69; Z=2.36, p=0.02). However, there was a large and statistically significant degree of heterogeneity between studies ($X^2$=12.0; df=3; p=0.007; $I^2$=75%). Given the high heterogeneity and inclusion of less than 10 studies in the meta-analysis, we did not have sufficient power to examine publication bias.[37] Two out of four studies were of high quality, scoring 9 on the Newcastle-Ottawa Scale, and examined large sample sizes[33 36] (see table 4). However, in each of the four meta-analysed studies, different assay methods were used and sampling methods varied significantly (see table 2). The lowest quality study to be included in the meta-analysis did not report mental health comorbidities and recruited participants from an inpatient setting.[34] There was also a wide range in the age of participants included in the four studies, with the largest study examining CRP levels in adolescents, which would likely contribute to high heterogeneity.

Five studies[33–36 38] (n=4669), one of which was conducted in children aged 16 years old[33] and two in participants with comorbid heart disease,[38] reported significantly higher CRP levels in participants with a diagnosis of GAD. The largest study[39] (n=5810) examining CRP in GAD examined CRP levels in children from baseline measurement aged 9–16 years to follow-up at age 19–21. This was the only study to examine the longitudinal association between GAD and CRP, and found a bivariate association both cross-sectionally and over time between GAD and elevated CRP; however, this was accounted for by potential covariates, including BMI and medication use. The only study[40] to find an inverse correlation between CRP and GAD was conducted in non-smoking women from a study in Finland and did not specify the numbers of participants with a diagnosis of GAD or group differences.

No difference was found in a cohort study (n=821) that used a combined inflammatory index consisting of CRP, IL-6 and TNF-α in 93 patients with a diagnosis of GAD and controls with a history of CVD.[41] Subgroup analysis examining differences in individual inflammatory markers was not reported.[41]

We found two studies[35 42] (n=196) that examined the association of severity of GAD symptoms with CRP level.

**Table 3** Summary inflammatory marker findings in GAD

| Study | N Controls | With GAD (n) | Findings |
|---|---|---|---|
| | | | *CRP* |
| Bankier et al[38] | 30 | 15 | ↑ in GAD with comorbid CVD compared with controls using a dichotomous outcome of CRP cut-off score (CRP >3 mg/L). |
| Copeland et al[39] | 5664 | 146 | Longitudinal study in adolescents: ↑ bivariate association both cross-sectionally and over time between GAD and elevated CRP, but accounted for by medication use and BMI. |
| De Berardis et al[42] | No control | 70 | ↑ in patients with GAD with comorbid alexithymia and with increased suicidal ideation, no control group. |
| Khandaker et al[33] | 3392 | 26 | ↑ in children aged 16 years old with GAD compared with controls, remained ↑ after adjusting for covariates. |
| Korkeila et al[40] | 116 | | ↓ in non-smoking women with a diagnosis of GAD compared with controls; however, control group was not described. |
| Nayek and Ghosh[34] | 50 | 50 | ↑ in patients with GAD compared with controls. |
| Tang et al[35] | 48 | 48 | ↑ in patients with GAD compared with controls and ↑ with increased severity of GAD. |
| Vogelzangs et al[36] | 556 | 454 | ↑ in patients with GAD compared with controls in unadjusted data obtained from the authors. |
| Zahm[41] | 728 | 93 | ↔ between those with and without a current GAD diagnosis (p=0.28) or with and without a lifetime GAD diagnosis, using a combined inflammatory index of CRP, IL-6 and TNF-α measurements. |
| | | | *IL-1* |
| Yang et al[45] | 41 | 28 | ↑ sputum IL-1 in patients aged 50–60 years old with GAD compared with controls. |
| | | | *IL-1α* |
| Tang et al[35] | 48 | 48 | ↑ IL-1α in patients with GAD compared with controls and ↑ with increased severity of GAD. |
| | | | *IL-2* |
| Tang et al[35] | 48 | 48 | ↑ in patients with GAD compared with controls (p<0.001) but ↔ with severity of GAD. |
| Tofani et al[46] | 10 | 14 | ↔ in patients with GAD compared with controls. |
| | | | *IL-4* |
| Hou et al[47] | 64 | 54 | ↔ in patients with GAD compared with controls. |
| | | | *IL-5* |
| Tang et al[35] | 48 | 48 | ↔ in patients with GAD compared with controls, or association with severity of GAD. |
| | | | *IL-6* |
| Hoge et al[44] | – | 70 | No control group: RCT of psychological intervention in GAD. |
| Tang et al[35] | 48 | 48 | ↑ in patients with GAD compared with controls and ↑ with increased severity of GAD. |
| Vogelzangs et al[36] | 556 | 454 | ↑ in patients with GAD compared with controls in unadjusted data obtained from author, but ↔ between IL-6 and GAD compared with other anxiety disorders. |
| Yang et al[45] | 41 | 28 | ↑ sputum in patients with GAD aged 50–60 years old compared with controls. |
| Zahm[41] | 728 | 93 | ↔ between those with and without a current GAD diagnosis or with and without a lifetime GAD diagnosis, using a combined inflammatory index of CRP, IL-6 and TNF-α measurements. |
| | | | *IL-8* |
| Tang et al[35] | 48 | 48 | ↑ in patients with GAD compared with controls and ↑ with increased severity of GAD. |
| | | | *IL-10* |
| Hou et al[47] | 64 | 54 | ↓ in patients with GAD compared with controls, which remained ↓ after adjustment for covariates. |

**Table 3** Continued

| Study | N Controls | With GAD (n) | Findings |
|---|---|---|---|
| Tofani et al[46] | 10 | 14 | ↑ in GAD compared with controls. |
| | | *IL-12p70* | |
| Tang et al[35] | 48 | 48 | ↑ in patients with GAD compared with controls but ↔ with severity of GAD. |
| | | *IFN-γ* | |
| Hou et al[47] | 64 | 54 | ↑ in patients with GAD compared with controls which remained ↑ after adjustment for covariates. |
| Korkeila et al[40] | 116 | | ↓ in non-smoking women with a diagnosis of GAD compared with controls; however, control group was not described. |
| Tang et al 2017[35] | 48 | 48 | ↑ in patients with GAD compared with controls and ↑ with increased severity of GAD. |
| | | *TNF-α* | |
| Hoge et al[44] | – | 70 | No control group: RCT of psychological intervention in GAD. |
| Hou et al 2017[47] | 64 | 54 | ↑ in patients with GAD compared with controls which remained ↑ after adjustment for covariates. |
| Korkeila et al[40] | 116 | | ↑ in non-smoking women with a diagnosis of GAD compared with controls, although control group was not described. |
| Vogelzangs et al[36] | 556 | 454 | ↔ in patients with GAD compared with controls, and ↔ between TNF-α and GAD compared with other anxiety disorders. |
| Yang et al[45] | 41 | 28 | ↑ sputum TNF-α in patients with GAD aged 50–60 years old compared with controls. |
| Zahm[41] | 728 | 93 | ↔ between those with and without a current GAD diagnosis or with and without a lifetime GAD diagnosis, using a combined inflammatory index of CRP, IL-6 and TNF-α measurements. |
| | | *CCL-5/RANTES* | |
| Ogłodek et al[48] | 40 | 120 | ↑ in men with GAD and comorbid personality disorder compared with controls. |
| | | *MCP-1* | |
| Ogłodek et al[48] | 40 | 120 | ↑ in GAD and comorbid personality disorder compared with controls. |
| | | *SDF-1* | |
| Ogłodek et al[48] | 40 | 120 | ↑ in GAD and comorbid personality disorder compared with controls. |
| | | *GM-CSF* | |
| Tang et al[35] | 48 | 48 | ↑ in patients with GAD compared with controls and ↑ with increased severity of GAD. |

↑,statistically significant increase in inflammatory marker in people with GAD compared with controls (p<0.05); ↓,statistically significant decrease in inflammatory marker in people with GAD compared with controls (p<0.05); ↔,no statistically significant difference in inflammatory marker in people with GAD compared with controls (p>0.05).
BMI, body mass index; CCL-5, chemokine C-C motif ligand 5; CRP, C reactive protein; CVD, cardiovascular disease; GAD, generalised anxiety disorder; GM-CSF, granulocyte-macrophage colony-stimulating factor; IFN-γ, interferon-γ; IL, interleukin; MCP-1, monocyte chemoattractant protein-1; RANTES, regulated on activation, normal T cell expressed and secreted; RCT, randomised controlled trial; SDF-1, stromal derived factor-1; TNF-α, tumour necrosis factor-α.

One found a significant positive correlation between CRP level and Generalised Anxiety Disorder Assessment (GAD-7) scores,[35] and the other reporting CRP differences in 70 patients with GAD with and without a diagnosis of alexithymia found a significant association between higher CRP and suicidal ideation.[42]

Although the meta-analysis and the majority of included studies reported raised CRP in GAD, there was wide variation in reporting and adjustment for important potential moderators, including comorbid MDD, use of medications, assay used and time of day of blood collection, all of which likely contributed to the high degree of heterogeneity between studies. Of the nine studies to analyse CRP, four (44.4%) did not exclude or adjust for medication use by participants.[33 38 40 41] Comorbid MDD was not adjusted for in the analysis by two studies,[39 41] one of which was included in the meta-analysis.[39] Only three of the nine studies reported time of sample collection[35 36 42] or whether this was in a fasted state,[36 41 42] and although all studies used a similar assay method, different assay types were used in every study.

**Table 4** Study quality assessment: Newcastle–Ottawa Scale

| | Selection | | | | Comparability | Exposure | | | |
| | Adequate case definition | Cases representative | Selection of controls | Definition of controls | Comparability of design and analysis | Ascertainment of exposure | Same method of ascertainment | Non-response rate | Total stars |
|---|---|---|---|---|---|---|---|---|---|
| Bankier et al[38] | ◇ | – | ◇ | ◇ | ◇◇ | ◇ | ◇ | ◇ | 8 |
| Copeland et al[39] | ◇ | ◇ | ◇ | ◇ | ◇◇ | ◇ | ◇ | – | 8 |
| De Berardis et al[42] | ◇ | – | NA | NA | NA | ◇ | NA | – | 2 |
| Hoge et al[44] | ◇ | – | NA | NA | NA | ◇ | NA | – | 2 |
| Hou et al[47] | ◇ | ◇ | ◇ | ◇ | ◇◇ | ◇ | ◇ | – | 8 |
| Khandaker et al[33] | ◇ | ◇ | ◇ | ◇ | ◇◇ | ◇ | ◇ | ◇ | 9 |
| Korkeila et al[40] | ◇ | – | ◇ | – | ◇ | – | ◇ | – | 4 |
| Nayek and Ghosh[34] | ◇ | – | ◇ | – | ◇◇ | ◇ | ◇ | – | 6 |
| Ogtodek et al[48] | ◇ | – | ◇ | ◇ | ◇◇ | – | ◇ | – | 6 |
| Tang et al[35] | ◇ | ◇ | ◇ | ◇ | ◇◇ | ◇ | ◇ | – | 8 |
| Tofani et al[46] | ◇ | – | – | – | ◇ | ◇ | ◇ | – | 4 |
| Vogelzangs et al[36] | ◇ | ◇ | ◇ | ◇ | ◇◇ | ◇ | ◇ | ◇ | 9 |
| Yang et al[45] | ◇ | ◇ | – | ◇ | ◇◇ | ◇ | ◇ | – | 7 |
| Zahm[41] | ◇ | – | – | – | ◇◇ | ◇ | ◇ | – | 5 |

–, did not meet the criteria; ◇, met criteria for allocation of point on Newcastle–Ottawa scale; ◇◇, two points on Newcastle–Ottawa scale; NA, not applicable.

**Table 5** Additional critical appraisal

| | Type of publication | Unrepresentative recruitment methods | Unrepresentative demographics | Between-group differences reported | Adjusted for between-group differences |
|---|---|---|---|---|---|
| Bankier et al[38] | Paper | Yes, recruited from cardiology clinic | Yes, older cohort due to cardiac comorbidity required | NR | NR |
| Copeland et al[39] | Paper | No | Yes, aged 9–21 only | Yes | Yes |
| De Berardis et al[42] | Paper | No | No (aged 18–45) | No control | No control |
| Hoge et al[44] | Paper | Yes, recruited by advert as part of parent RCT | No (aged >18) | No control | No control |
| Hou et al[47] | Paper | No | No (aged 18–65) | Yes | Yes |
| Khandaker et al[33] | Paper | No | Yes, aged 16 years old only | Yes | Yes |
| Korkeila et al[40] | Abstract | Yes, recruited from existing study in Finland | Yes, non-smoking women only | No | Yes (BMI only) |
| Nayek and Ghosh[34] | Paper | Yes, inpatients only | No (aged 18–65) | Yes | Yes |
| Ogłodek et al[48] | Paper | Yes, comorbid personality disorder | No | Yes | No |
| Tang et al[35] | Paper | No | No (aged 18–60) | Yes | Yes |
| Tofani et al[46] | Abstract | Recruitment method not stated | NR | NR | NR |
| Vogelzangs et al[36] | Paper | No | No (aged 18–65) | Yes | Yes |
| Yang et al[45] | Paper | No | Yes (aged 50–60) | Yes | Yes |
| Zahm[41] | Dissertation | Yes, cardiology patients only | Yes (aged>50) | Yes | Yes |

BMI, body mass index; NR, not reported; RCT, randomised controlled trial.

In summary, of the nine studies to have examined differences between GAD and controls, the majority reported raised CRP in GAD and the meta-analysis found significantly raised CRP in GAD with a small effect size. However, there was a wide variation in study methods, including variable adjustment for mediators of inflammation such as comorbid MDD, medication use and sampling methods. Only one study examined CRP in GAD longitudinally, reporting a bivariate association accounted for by health-seeking behaviours.[39]

## Interleukins

Seven studies examined the association between interleukins and GAD (see table 3). IL-6 is a mediator of T cell and B cell activation and induces acute phase proteins in hepatocytes, among other functions.[32] Pharmacological blockade of IL-6 action is used to treat several autoimmune conditions including RA, and raised IL-6 has been associated with a number of psychiatric conditions including depression, schizophrenia and post-traumatic

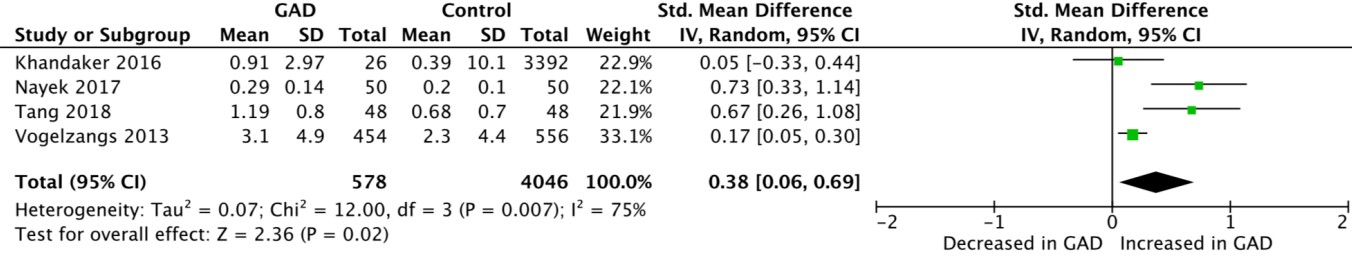

**Figure 2** Random-effects meta-analysis of CRP levels in GAD versus controls. CRP, C reactive protein; GAD, generalised anxiety disorder; Std, standard; IV, inverse variance.

stress disorder (PTSD).[32 43] We found IL-6 was the most frequently measured interleukin, with five studies (n=2066) examining changes in patients with GAD compared with controls.[35 36 41 44 45] The largest study investigated differences between 454 participants with a diagnosis of GAD and 556 controls from The Netherlands Study of Depression and Anxiety cohort.[36] Although analysis was conducted on anxiety disorders as a whole, the mean difference in IL-6 in people with GAD compared with controls obtained through direct communication with the author showed significantly higher levels in GAD. However, it is unclear whether these differences remain significant after adjustment for group differences and no associations were found between IL-6 and participants who had all types of anxiety disorder.[36] Two studies[35 45] (n=165), one of which used saliva samples,[45] reported significantly higher IL-6 in medication-naïve participants with a diagnosis of GAD compared with age-matched and sex-matched healthy controls.

No difference was found in a combined inflammatory index consisting of CRP, IL-6 and TNF-α in a study of 93 patients with GAD and comorbid ischaemic heart disease.[41] One case-controlled study[35] of 48 Chinese outpatients presenting for the first time with a diagnosis of GAD and 48 age-matched, sex-matched and education-matched controls accounted for all results for IL-1α, IL-5, IL-8 and IL-12p70. This study found significantly higher levels of IL-1α, IL-8 and IL-12p70 in patients with GAD, in addition to higher levels of IL-1α and IL-8 with increased severity of GAD (as measured by the GAD-7 scale), but did not account for chronic physical comorbidities during recruitment or in analysis. Both IL-1α and IL-8 have proinflammatory functions as chemoattractants for leukocytes and haematopoiesis, and have been targeted for treatments in a number of autoimmune conditions.[32] However, there was no association between GAD and IL-5, which is thought to predominantly mediate myeloid cell activation and is a target of treatment in asthma.[32] The same study[35] also examined IL-2, which has a major role in T cell-mediated autoimmune and inflammatory conditions.[32] The results showed significantly higher IL-2 in patients with GAD; however, this conflicted with results from a smaller study (n=24) that found no significant difference between medication-naïve patients with GAD and controls, although few details of the characteristics of participants, sampling or analysis were reported in this abstract.[46]

One study that measured IL-1 using sputum analysis found significantly higher levels in GAD compared with controls in 69 participants recruited from the same Chinese hospital.[45] Although IL-1 is proinflammatory, there are differences in function, dependent on the class of IL-1 protein measured which was not reported in this study.[45]

IL-4 has several proinflammatory functions, including IgE class switching, expression of major histocompatibility complex (MHC) class II and acts as a survival factor for T and B cells.[32] The only study to measure IL-4 found

no differences between 54 patients with GAD recruited from community mental health teams and primary care after controlling for age, sex, BMI, smoking, alcohol consumption and comorbid depression.[47] This study[47] also investigated IL-10, which was the only cytokine with an anti-inflammatory function to be measured and is involved in immunosuppression of T cell subsets and B cell immunoglobulin production. This found significantly lower levels of IL-10 (OR 0.35, p=0.003) in patients with GAD. However, this opposed findings from a smaller study (n=24) which reported significantly higher levels of IL-10 in patients with GAD compared with controls, although it was not reported whether this association remained significant after controlling for group differences.[46]

In summary, IL-6 was the most commonly measured interleukin raised in GAD compared with controls in the majority of studies; however, no study examined the longitudinal association with GAD. Other interleukins were examined by relatively few studies that examined small numbers cross-sectionally with mixed findings.

### Interferon-γ
IFN-γ has antiviral roles, including promoting cytotoxic activity, MHC class I and II upregulation, natural killer (NK) cell activation, and is a treatment target in inflammatory conditions such as Crohn's disease.[32] Three studies investigated IFN-γ levels in GAD (n=330).[35 40 47] The largest study (n=118) found higher IFN-γ in patients with GAD from the UK that remained significant after adjustment for age, gender, BMI, smoking, alcohol and comorbid depression, but did not adjust for anxiolytic medication use in analysis.[47] This finding was supported by a study of 96 participants which reported higher IFN-γ levels in GAD and a significant positive correlation between anxiety severity and IFN-γ.[35] Conflicting findings were reported by a Finnish study of 116 participants, which found significantly lower IFN-γ in patients with GAD. However, the number of participants with a diagnosis of GAD, differences between groups and adjustment for potential confounders were not reported.[40] In summary, only a few small cross-sectional studies have examined differences in IFN-γ between GAD and control groups, and their findings were mixed.

### Tumour necrosis factor-α
TNF-α has a wide array of roles in host defence, including initiating a strong acute inflammatory response but limiting duration of inflammatory activation, and is the target of blocking monoclonal antibodies in the treatment of a wide array of autoimmune conditions including Crohn's disease and RA.[32] Six studies (n=2300) investigated TNF-α in GAD, with mixed findings. Three studies (n=303) found TNF-α significantly raised in patients with GAD compared with controls.[40 45 47] However, the largest study to measure TNF-α (n=1010) found no difference between participants with GAD and controls, and no correlation between TNF-α and anxiety symptoms.[36] This finding was supported by a study of 93 patients

with GAD and comorbid ischaemic heart disease using a combined inflammatory index of CRP, IL-6 and TNF-α which reported no difference compared with controls.[41] In summary, although the majority of the studies to measure differences in TNF-α between GAD and controls reported significantly raised levels, these comprised small cross-sectional studies and the largest study reported no difference.

### Other cytokines
One study compared the levels of the proinflammatory cytokines chemokine C-C motif ligand 5 (CCL-5), monocyte chemoattractant protein-1 (MCP-1) and stromal derived factor-1 (SDF-1) in 120 medication-naïve, physically well patients with a diagnosis of GAD and comorbid personality disorder with 40 controls.[48] Significantly higher levels of MCP-1 and SDF-1 were reported in both men and women, and higher CCL-5 in men but not women with a diagnosis of GAD compared with controls.[48]

### DISCUSSION
To our knowledge, this is the first systematic review and meta-analysis focusing on inflammatory cytokines in GAD. Using a range of databases we identified 14 studies, comprising 1188 participants with GAD and which measured 16 cytokines. We found significantly raised levels of CRP, IFN-γ and TNF-α in people with GAD compared with controls, which were findings replicated in two or more studies. A further 10 proinflammatory cytokines were reported to be significantly raised in GAD in at least one study; however, 6 of 14 studies found no difference in at least one cytokine.

Despite substantial efforts to acquire data by contacting the authors, it was only possible to conduct a meta-analysis of CRP. This identified significantly higher levels in GAD compared with controls with a small effect size (SMD=0.38), although there was evidence of significant heterogeneity across studies ($I^2$=75%). This effect size in CRP is greater than has been reported in other anxiety disorders (PTSD: SMD=−0.14[24] or MDD (SMD=0.14[6]) and is similar to that reported in schizophrenia (SMD=0.45).[49] However, the effect size of our meta-analysis was driven by findings in poorer quality studies with small sample sizes. The two higher quality, larger studies reported a smaller effect size and no significant difference between groups, respectively. As a result, further high-quality studies are required to confirm our findings of raised CRP in GAD.

Although we were only able to meta-analyse CRP, meta-analyses of different cytokines in other anxiety disorders have been conducted with larger effect sizes. A meta-analysis of inflammatory markers in PTSD identified 20 studies which reported increased IL-6, IL-1β, TNF-α and IFN-γ levels, with effect sizes ranging from small (IFN-γ: SMD 0.49) to large 1.42 (IL-1β: SMD 1.42).[24] However, a systematic review and meta-analysis of proinflammatory cytokines in OCD identified 12 studies,

and concluded that there was a significant reduction in IL-1β with moderate effect size (SMD=−0.60, p<0.001), and only IL-6 levels were significantly increased after subgroup analysis in medication-free adults with OCD.[7] It is unclear whether this profile of inflammatory marker changes would follow a similar pattern in GAD if future studies enabled further meta-analysis.

In light of the high heterogeneity among studies, low participant numbers, and inconsistent reporting and adjustment for known confounding factors such as BMI, smoking, medication use and comorbidities, our findings should be interpreted with caution. It was not possible to analyse the cause of the degree of heterogeneity due to the paucity of studies. Other known mediators of inflammation[24] such as physical activity, raised blood pressure and genetics were not accounted for. Furthermore, reporting of GAD severity and duration of symptoms was generally poor, preventing detailed analysis of whether inflammatory markers predicted outcomes and quality of life. We also found limitations in inclusion of specific demographics of participants with GAD. For example, despite GAD in older adults being prevalent and often treatment-resistant,[50–52] only two studies included participants over the age of 65, both of which only included patients with comorbid ischaemic heart disease.

We are beginning to understand the interplay between cytokines, the immune system and mental health.[1 53] At a molecular level we are aware that proinflammatory cytokines, including IFN, IL-1β and TNF, can reduce the availability of monoamines by inducing expression of presynaptic reuptake pumps and inhibiting enzymes involved in monoamine synthesis,[54] linking the monoamine theory of anxiety with inflammatory mechanisms. There is also a growing understanding of the relationship between systemic inflammation and the central nervous system (CNS).[1 55] Microglial activation has been shown to be mediated by peripheral cytokines, and increased activation has been found in postmortem studies of patients with MDD and schizophrenia.[1] No study we identified correlated inflammatory marker changes with in vivo microglial activation imaging in GAD, and to our knowledge no research on postmortem microglial changes in GAD has been conducted. Increased neuronal activity has also been shown to induce inflammatory and vascular changes in the brain, suggesting that psychological stress can not only be induced by inflammation but perpetuate chronic low-grade inflammation seen in other vascular and neurodegenerative disorders.[55] Understanding interactions between the CNS and immune system and identifying biomarkers of GAD offer potential for novel therapeutic approaches. The revolution of development of monoclonal-antibody therapies for inflammatory disorders[56] raises the possibility of repurposing these medications for trials in treatment-resistant GAD if specific and consistent profiles of inflammatory biomarkers are identified.

However, it remains unclear as to whether inflammation plays a causal role in GAD.[43 54] For example, although

IL-6 is a successful target for treatment in a number of autoimmune conditions and raised IL-6 is implicated in several psychiatric disorders, it also acts to reduce other proinflammatory cytokines such as TNF via negative feedback and is induced by physical exercise, hyperthermia, fasting, sleep deprivation and sunlight exposure without activation of other proinflammatory cytokines.[43] This raises the question as to whether inflammation in GAD is a consequence rather than a cause of symptoms. This will only be answered by large prospective longitudinal studies, better characterising the relationship between inflammation and GAD. However, remarkably our review identified only one longitudinal study of inflammation in patients with GAD that examined a cohort of adolescents until the age of 21 and only investigated CRP.

Recent studies using Mendelian randomisation in depression have suggested that cytokines such as IL-6 are causal risk factors for depression,[57] and trials of immunotherapy in psychosis are already under way.[58] Our study suggests that GAD is an important candidate for future similar studies exploring causality of inflammation and potentially novel drug trials.

## CONCLUSION

There is some preliminary evidence to suggest a raised inflammatory response in GAD, although it is unclear whether inflammatory cytokines play a role in the aetiology. GAD remains a poorly studied area of psychiatric neuroinflammatory research compared with other mental illnesses such as MDD and schizophrenia. While we are a long way from using inflammatory cytokines as a biomarker or treatment target in GAD, current findings reflect inflammatory changes seen in other mental illnesses and highlight the importance of ongoing investigation of the role inflammation plays in the development and course of GAD. Further, methodologically consistent, prospective, longitudinal studies examining the mechanisms and relationship between inflammation and GAD, while accounting for known mediators of cytokine production, are required.

**Contributors** RH, RLG and HC were involved in the initial design of the research. EA and HC performed the data extraction of the included studies and quality analysis. HC wrote the initial draft of the manuscript and did the statistical analysis, with supervision from RLG and RH. RH, RLG and HC participated in the critical revision of the article, and all authors approved the final article.

**Funding** HC and EA are NIHR Academic Clinical Fellows (ACF). This research was supported by the NIHR Biomedical Research Centre at University College London/ University College Hospital London.

**Competing interests** None declared.

**Patient consent for publication** Not required.

**Provenance and peer review** Not commissioned; externally peer reviewed.

**Data sharing statement** All data extracted for this systematic review and meta-analysis are available via direct contact with HC.

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
