## [Reviewer comments · BMJ Open]

ARTICLE DETAILS

TITLE (PROVISIONAL)	A systematic review and meta-analysis of the association between peripheral inflammatory cytokines and generalised anxiety disorder
AUTHORS	Costello, Harry; Gould, Rebecca; Abrol, Esha; Howard, R

VERSION 1 - REVIEW

REVIEWER	Wedekind; Dirk University of Goettingen, Dept. of Psychiatry and Psychotherapy von Siebold Strasse 5, 37075 Goettingen, Germany
REVIEW RETURNED	11-Dec-2018

GENERAL COMMENTS	Costello H, et al: A systematic review and meta-analysis of the association between peripheral inflammatory cytokines and generalised anxiety disorder The authors contribute a high quality meta-analysis on a timely and relevant issue. The importance of inflammatory processes in psychiatric disorders is increasingly recognized; nevertheless it has been focused by research for more than two decades. The amount of studies is fairly larger for other major disorders than GAD, such as major depression or schizophrenia. Yet, the importance for anxiety disorders appears to be striking, the methods are sound and the presentation of results is appealing. This manuscript may be recommended for publication in BMJ Open after some minor issues have been settled. The authors say that no longitudinal studies have been performed for the majority of biomarkers. This is remarkable since changes of GAD severity might well be associated with immune system markers. The authors would do well on commenting on this issue in the discussion or introductory part. The literature in this respect gives the impression that some studies did not elaborate the background of the chosen biomarkers very well. There is a large heterogeneity across studies and preliminary evidence might exist for only few of the investigated markers. To give the reader a comprehensive idea of what these findings could mean for the pathogenesis of GAD I would appreciate to see some implications in the discussion part and possible relationships to the finding in MD or schizophrenia.
--

REVIEWER	Audrey Rankin Queens University Belfast, N Ireland
REVIEW RETURNED	21-Jan-2019

GENERAL COMMENTS	This paper reports a meta-analysis which estimates the association between peripheral inflammatory cytokines and generalised anxiety disorder. The statistical analysis conducted utilises the standardised mean difference (SMD) approach which is an appropriate method given the different measurement methods used. The only comment I have is in relation the quality assessment conducted according to the Newcastle-Ottawa (NOS) criteria. Although the authors have commented on the overall quality of studies included in the review, the authors could comment on the quality of the four studies included in the meta-analyses in terms of the certainty of the evidence and implications for the results/conclusions.
--

VERSION 1 – AUTHOR RESPONSE

Reviewer 1:

- The authors contribute a high quality meta-analysis on a timely and relevant issue. The importance of inflammatory processes in psychiatric disorders is increasingly recognized; nevertheless it has been focused by research for more than two decades. The amount of studies is fairly larger for other major disorders than GAD, such as major depression or schizophrenia. Yet, the importance for anxiety disorders appears to be striking, the methods are sound and the presentation of results is appealing. This manuscript may be recommended for publication in BMJ Open after some minor issues have been settled.

We thank the reviewer for their comments and are pleased that they consider the manuscript should be recommended for publication in BMJ Open (subject to revisions).

- The authors say that no longitudinal studies have been performed for the majority of biomarkers. This is remarkable since changes of GAD severity might well be associated with immune system markers. The authors would do well on commenting on this issue in the discussion or introductory part. The literature in this respect gives the impression that some studies did not elaborate the background of the chosen biomarkers very well.

We have now commented further on the lack of longitudinal studies in the discussion (page 12).

- There is a large heterogeneity across studies and preliminary evidence might exist for only few of the investigated markers. To give the reader a comprehensive idea of what these findings could mean for the pathogenesis of GAD I would appreciate to see some implications in the discussion part and possible relationships to the finding in MD or schizophrenia.

We have now commented on these implications in the discussion, particularly regarding future direction of research in GAD with regard to current research in MD and schizophrenia (page 12).

Reviewer 2:

- The only comment I have is in relation the quality assessment conducted according to the Newcastle-Ottawa (NOS) criteria. Although the authors have commented on the overall quality of studies included in the review, the authors could comment on the quality of the four studies included in the meta-analyses in terms of the certainty of the evidence and implications for the results/conclusions.

We have now commented specifically on the quality of studies included in the meta-analysis in the Results (page 7) and Discussion (page 10).

VERSION 2 – REVIEW

REVIEWER	Dirk Wedekind University of Goettingen, Dept. of Psychiatry and Psychotherapy, Goettingen, Germany
REVIEW RETURNED	29-Apr-2019

GENERAL COMMENTS	According to the minor issues I had for the previous Version, these now appear to be well solved. I may recommend the manuscript for publication in BMJ in the present form.
--

REVIEWER	AUDREY RANKIN Queens University Belfast
REVIEW RETURNED	19-Apr-2019

GENERAL COMMENTS	Thank you for the opportunity to review the revised manuscript. It is clear that the authors have made a substantial effort in revising this paper. Regarding my comments, I am happy with the changes and/or responses given
---